Gammaretroviral vector encoding a fluorescent marker to facilitate detection of reprogrammed human fibroblasts during iPSC generation

Srinivasakumar Narasimhachar 1 skumar15@slu.edu
Zaboikin Michail 1
Tidball Andrew M. 2
Aboud Asad A. 2
Neely M. Diana 2
Ess Kevin C. 2
Bowman Aaron B. 2
Schuening Friedrich G. 1
1 Division of Hematology/Oncology, Department of Internal Medicine, Saint Louis University , Saint Louis, MO , USA
2 Vanderbilt University Medical Center, Department of Neurology, Vanderbilt Kennedy Center for Research on Human Development , Nashville, TN , USA
Rehman Jalees
Electronic publication date: 2013 Dec 10
Publication date: 2013
Volume: 1
Electronic Location ID: e224
Received 2013 Jun 27; Accepted 2013 Nov 22
Copyright: © 2013 Srinivasakumar et al.
Copyright year: 2013
Copyright holder: Srinivasakumar et al.
License: This is an open access article distributed under the terms of the Creative Commons Attribution License, which permits unrestricted use, distribution, and reproduction in any medium, provided the original author and source are credited.
License URL: https://creativecommons.org/licenses/by/3.0/

Keywords: Gammaretrovirus, Induced pluripotent stem cells, Reprogramming, IPSC, Detection, Silencing

Funding: NIH/NIEHS RO1 ES016931 Doris Duke Charitable Foundation and NINDS, NIH R01NS078289 ABB was supported by NIH/NIEHS RO1 ES016931. KCE was supported by the Doris Duke Charitable Foundation and NINDS, NIH R01NS078289. FGS, KCE and ABB were also supported by departmental and institutional funds. The funders had no role in study design, data collection and analysis, decision to publish, or preparation of the manuscript.

==============================
Induced pluripotent stem cells (iPSCs) are becoming mainstream tools to study mechanisms of development and disease. They have a broad range of applications in understanding disease processes, in vitro testing of novel therapies, and potential utility in regenerative medicine. Although the techniques for generating iPSCs are becoming more straightforward, scientists can expend considerable resources and time to establish this technology. A major hurdle is the accurate determination of valid iPSC-like colonies that can be selected for further cloning and characterization. In this study, we describe the use of a gammaretroviral vector encoding a fluorescent marker, mRFP1, to not only monitor the efficiency of initial transduction but also to identify putative iPSC colonies through silencing of mRFP1 gene as a consequence of successful reprogramming.

Introduction

Induced pluripotent stem cells (iPSCs) have many properties of embryonic stem cells and, therefore, hold great promise for their widespread utility in understanding stem cell biology and furthering regenerative medicine (Yamanaka, 2012). Retroviral and Epstein-Barr virus based vectors encoding either the “Yamanaka” (Oct4, Klf4, Sox2, c-Myc or L-Myc, Lin28, and anti-p53 shRNA) (Okita et al., 2011; Takahashi et al., 2007) or “Thompson” (Oct4, Sox2, Nanog, Lin28) (Yu et al., 2009; Yu et al., 2007) reprogramming factors (RFs) can be used to generate iPSCs. Alternative methodologies to generate iPSCs using proteins, nonviral minicircle vectors, synthetic modified RNA, or small molecules have also been described (Jia et al., 2010; Kim et al., 2009; Warren et al., 2010; Yu et al., 2011; Zhou et al., 2009).

iPSC reprogramming appears to be a consequence of changes in DNA methylation patterns and histone modifications resulting in chromatin remodeling (Liang & Zhang, 2013). Previously active regions responsible for the differentiated cellular phenotype are silenced while other regions are concomitantly activated to yield a gene expression pattern similar to that seen in embryonic stem cells (Maherali et al., 2007; Stadtfeld et al., 2008). The reprogramming is initiated by transient expression of RFs and follows an orderly process delineated during derivation of murine iPSCs (Stadtfeld et al., 2008). Alkaline phosphatase activation is an early discernable event followed by silencing of fibroblast specific genes such (e.g., THY1, COL5A2, FBN2). Expression of endogenous stem cell genes including SSEA-1 appears. Finally, silencing of retroviral vector-derived gene expression is seen in conjunction with activation of endogenous OCT4 and NANOG genes. More recent investigations reveal that initial stochastic gene expression patterns, following initiation by the RFs, precede the more orderly and deterministic expression patterns identified at subsequent stages of reprogramming (Buganim, 2012 #5127; Hanna et al., 2009).

While the techniques for generating iPSCs might appear simple, accurate identification of fully reprogrammed iPSC colonies can prove difficult. The observation that successful reprogramming of cells to iPSC-like state is associated with a loss of expression of genes under control of the retroviral long terminal repeat (LTR) promoter enabled us to exploit this feature for identification of promising iPSC colonies. Here, we describe the use of a gammaretroviral vector encoding a fluorescent marker for not only ensuring adequate transduction efficiency of fibroblasts but also to identifying putative iPSC colonies based on silencing of the mRFP1 marker gene.

Materials and Methods

Cells

Human embryonic kidney 293T (HEK293T) cells were obtained from American Type Culture Collection (ATTC; catalog number SD-3515) and maintained in Dulbecco’s modified Eagle’s medium containing 2 mM L-glutamine, 100 U/ml of penicillin, 100 µg/ml streptomycin and 10% heat-inactivated fetal bovine serum (FBS) (Hyclone/ThermoFisherScientific, USA). Human lung fibroblasts were obtained from ATCC (MRC-5, catalog number CCL-171) and maintained in Eagle’s minimal essential medium containing 10% FBS. Mouse ES feeder cells (SNL-76/7-4) that express leukemia inhibitory factor (LIF) and puromycin phosphotransferase were obtained from Wellcome Trust Sanger Institute and maintained in Knockout DMEM with 7% FBS, Penicillin (50 U/ml) and Streptomycin (50 µg/ml). The mouse feeder cells are also available from ATCC (catalog number SNLP 76/7-4).

Plasmids

The following plasmid vectors were obtained from Addgene.org: Plasmids pMXs-hOCT3/4 (catalog number 17217), pMXs-hSOX2 (catalog number 17218), pMXs-hKLF4 (catalog number 17219), and pMXs-hc-MYC (catalog number 17220) were made available by the Yamanaka Laboratory. Plasmids pUMVC (encodes Murine leukemia virus Gag/Pol, catalog number 8449), and pCMV-VSV-G (encodes VSV-G envelope, catalog number 8454) were made available by the Weinberg Laboratory. Plasmid pMXs-mRFP1 encodes monomeric red fluorescent protein (Hotta et al., 2009b) (catalog number 21315) and was made available by the Ellis Laboratory.

Production of vector stocks

Vector stocks were produced in HEK293T cells using CaPO4-mediated transient transfection protocol (Srinivasakumar, 2002). In preparation for transfection, T-75 cell culture flasks were seeded with 7.5 × 106 HEK293T cells on the previous day. The plasmid DNAs (7.5 µg of pUMVC, 0.6 µg of pCMV-VSV-G and 22.5 µg of the gene-transfer vector encoding the RF or mRFP1) were resuspended in 1.5 ml of CaCl2 solution (0.25 M). The DNA was precipitated by adding drop wise 1.5 ml of HEPES buffered saline, pH 7.05 (50 mM HEPES, 10 mM KCl, 12 mM dextrose, 280 mM NaCl, 1.5 mM Na2HPO4), while bubbling air through the DNA-CaCl2 solution. The mix was immediately distributed drop wise onto the HEK293T cells. The following day, the medium was replaced with fresh medium, and vector-containing supernatant was harvested 48 h later by centrifugation at 1,400 × g for 15 min at 4°C and stored at −80°C in aliquots. After determination of vector titers (see below), pools of required vectors were made to give the desired multiplicity of infection (MOI), and concentrated by ultracentrifugation at 100,000 × g for 2 h at 4°C. The pelleted vectors were resuspended in a minimal volume of complete MRC-5 growth medium for transduction of human fibroblasts (see below).

Determination of vector titers

HEK293T cells were seeded in 6-well plates (2.5 × 105 cells/well) the day prior to infection. The next day, an aliquot of the vector-containing supernatant was added to the cells in one ml of fresh medium containing 8 µg of polybrene. After overnight incubation, polybrene was diluted by adding an additional 2 ml of fresh medium. The following day, the cells were rinsed several times with PBS and released by trypsin-EDTA treatment. The cells were pelleted and washed with PBS prior to isolation of genomic DNA using DNeasy kits from Qiagen (Maryland, USA) using the manufacturer’s recommended protocol and included an RNAse I treatment step.

The vector and β-actin copy numbers in the isolated genomic DNA were determined using qPCR in a Bio-Rad MyiQ thermocycler. Each qPCR reaction was carried out in a 25 µL final volume of iQ SYBR Green Supermix containing 60 ng of genomic DNA and 200 nM of each primer. We used a two-step PCR with a 95°C for 20 s denaturation step and 63.1°C for 45 s annealing and extension step. A final melt-curve analysis was done with 0.5°C temperature increments.

Transduction of human lung fibroblasts

MRC-5 fibroblasts were seeded in 6-well plates (60,000 cells/well). The cells were transduced on two consecutive days with pooled vector stocks containing all four RF vectors and the mRFP1 vector at an MOI of 5 using the “spin-transduction” method. This was done by centrifuging the plates at 1,200 × g for 2 h at room temperature (24°C) after the addition of the vectors in 1 ml of MRC-5 medium containing 8 µg/ml of polybrene. The plates were returned to the incubator after adding an additional 1 ml of medium. Forty-eight hours after the second spin-infection, the fibroblasts were split at a ratio of 1:6 and maintained for 3 more days in MRC-5 medium before seeding for reprogramming on mouse embryonic feeder cells.

Preparation of mouse ES feeder cells

SNL 76/7-4 cells were grown to about 90% confluency, washed once with PBS and incubated in SNL growth medium with 10 µg/ml Mitomycin C (Sigma, Saint Louis, USA; Catalog number M4287) for 2 h at 37°C. The cells were washed three times in excess PBS, released with Trypsin-EDTA and frozen in aliquots and stored in liquid nitrogen.

Reprogramming of MRC-5 cells after transduction

In preparation for reprogramming, 6-well plates were coated with gelatin (0.1%, Millipore Corp., USA; Catalog number ES-006-B) overnight at 37°C. The next-day Mitomycin-treated mouse ES feeder cells (SNL 76/7-4) were seeded onto 6-well plates (500,000 cells/well). The following day, each well received 10,000 transduced MRC-5 fibroblasts in MRC-5 medium. The medium was changed the following day to human ES reprogramming medium (Knockout DMEM/F12 containing 20% Knockout serum replacer (KOSR), 100 µM non-essential amino acids, 100 µM 2-mercaptoethanol, 1 mM L-glutamine, 50 U/ml penicillin, 50 µg/ml streptomycin, and 10 ng/ml basic fibroblast growth factor (bFGF, Millipore, Massachusetts, USA). The medium was replaced every other day for about 10 days and then replaced daily for additional one to 2 weeks until putative iPSC-like colonies appeared and grew to a size that could be manually picked. The iPSC-like colonies were marked using an object marker (BioIndustrial Products, USA; Catalog number 14361) and picked using drawn out Pasteur pipets in a laminar floor hood under a stereomicroscope. Picked colonies were deposited on Mitomycin C-treated SNL feeder layer containing cells in either 24-well or 6-well plates. In later experiments we determined that we could directly transfer iPSC-like clones onto feeder-free Matrigel-coated (Beckton-Dickinson, USA) wells. The cells were gradually transitioned to a feeder-free defined medium (mTeSR1; STEMCELL Technologies, Vancouver, Canada), over a 4-day period by increasing its percentage by 25% each day (Neely et al., 2011). Individual clones were expanded and frozen down in aliquots using mFreSR (STEMCELL Technologies, Vancouver, Canada) as per the manufacturer’s recommended protocol.

Characterization of iPSCs

Alkaline phosphatase

The iPSC clones were stained for the alkaline phosphatase pluripotency marker using either fixed or live cultures. For fixed cultures we used a kit from Sigma (Saint Louis, Missouri, USA; Catalog number 85L1-1KT), while for unfixed cultures we used the Alkaline Phosphatase Live Stain (Molecular Probes/Life Technologies, Grand Island, NY, USA; Catalog number A14353) as per the manufacturers’ recommended protocols.

Immunofluorescence assay for stem-cell markers

The putative iPSC clones were assessed for the expression of SSEA3, SSEA4, Tra-160, Tra-1-81 surface antigens and Oct4 and Nanog nuclear transcription factors using primary and fluorescent-labeled secondary antibodies as described previously (Neely et al., 2011).

Reverse-transcriptase-qPCR (RT-qPCR)

Expression of pluripotency marker genes (NANOG and DNMT3B), and RFs (OCT4, SOX2, KLF4 and cMYC), was determined using RT-qPCR of total mRNA isolated from individual IPSC clones using the primers listed in Table 3. The endogenous RF mRNA (RFE) that originated from the cell was estimated using primers targeting either the 3′ or 5′ untranslated region. The total RF mRNA (RFT) was quantitated using primers situated in the coding exon and measured both endogenous and vector-originated mRNAs. Some of the primers used in this study were originally published by Chan and coworkers (Chan et al., 2009) and subsequently validated by us in other studies (Neely et al., 2011). The reverse transcription was done with Superscript III kit (Invitrogen/Life Technologies, Grand Island, NY, USA; Catalog number: 11752250) using random hexamers according to the recommended protocol. The following cycling parameters were used for qPCR: Incubation at 95°C for 10 min, followed by 40 two-step cycles each consisting of 95°C for 15 s (denaturation) and 60°C for 60 s (annealing and extension). The qPCR was accomplished using the Power SYBR Green Master Mix (Invitrogen/Life Technologies, Grand Island, NY, USA; Catalog number: 4367659) in a 7900HT PCR System (Applied Biosystems).

Methylation-sensitive restriction enzyme qPCR (MSRE-qPCR)

Genomic DNA (60 ng), isolated using DNeasy Kits, was digested with 5 units of SmaI (methylation sensitive) or MscI (methylation insensitive) or incubated in the absence of restriction enzyme (uncut) in the manufacturer supplied buffer (New England Biolabs, Massachusetts, USA) in a 50 µl reaction volume at 37°C for 4 h. The reaction was terminated by inactivating the enzymes at 80°C for 20 min. An aliquot of the digest (5 µl) containing 6 ng of genomic DNA was then used in qPCR using primers (SK160 and SK161, Table 1) targeting the 5′ Moloney murine leukemia virus LTR and 5′ untranslated region (UTR) (Fig. 8A). The PCR was carried out using SsoAdvanced SYBR Green Supermix (Bio-Rad, Hercules, CA, USA; Catalog no. 1725261) in a Bio-Rad CFX96 thermocycler. The PCR cycling parameters consisted of a two-step amplification cycle consisting of a denaturation step at 98°C for 10 s and an annealing and extension step at 63.1°C for 30 s for a total of 40 cycles. A final melt-curve analysis was done as described above for vector titer determination. The vector copies were normalized to β-actin copy numbers present in the same samples as described earlier for vector titer determination.

Table 1 Primers used in qPCR for quantitation of Moloney murine leukemia virus (Mo-MLV) and human β-actin copy numbers in transduced cells and iPSCs.

Laboratory
designation	Description
(nucleotide position)	Product
size	Sequence (5′ – 3′)	Parent GenBank
Accession No.	
Psi-2 S	Mo-MLV packaging signal, S (751-770)	231 bp	CAACCTTTAACGTCGGATGG	J02255.1	
Psi-2 AS	Mo-MLV packaging signal, AS (962-981)	GAGGTTCAAGGGGGAGAGAC	J02255.1	
SK106	Human β-actin, S (477-497)	250 bp	CATGTACGTTGCTATCCAGGC	NM_001101	
SK107	Human β-actin, AS (706-726)	CTCCTTAATGTCACGCACGAT	NM_001101	
SK160	Mo-MLV U3a, S (8219-8238)	337 bp	TCTGCTCCCCGAGCTCAATA	J02255.1	
SK161	Mo-MLV downstream of PBSb, AS (271-291)	GCTAACTAGTACCGACGCAGG	J02255.1	
Notes.

a U3, Unique 3′ region.

b PBS, glutamine t-RNA primer binding site.

S sense

AS antisense

Karyotyping

iPSC clones were karyotyped by Genetics Associates Inc., (Nashville, TN) or by the Cytogenetics Department at Saint Louis University Medical Center (Saint Louis University, Saint Louis, MO) using standard techniques.

Differentiation into ectodermal, mesodermal and endodermal elements through embryoid body (EB) formation (Ohnuki, Takahashi & Yamanaka, 2009)

Six-well plates containing iPSCs in log-phase of growth and showing large to medium sized colonies, were treated with Dispase (2 mg/ml in DMEM/F12, Invitrogen/Life Technologies, Grand Island, NY, USA) for 20 min at 37°C. The colonies were released by washing gently with the Dispase solution, and transferred to a 15 ml conical tube. The colonies were allowed to settle to the bottom of the tube at unit gravity, washed once by gently removing and replacing the supernatant with DMEM/F12, and then resuspended in EB formation medium (DMEM with 20% FBS or human ES reprogramming medium described in Materials and Methods but without bFGF). The colonies in EB formation medium were transferred to low-attachment T-25 flasks or 6-well plates (Corning, USA) and placed on an orbital shaker in 37°C humidified incubator. After 8 days, with alternate day medium change, the resultant EBs were transferred to 6-well plates coated with gelatin and allowed to attach and spread out. After additional 8 days, with regular medium change every other day, the cells were fixed using 4% paraformaldehyde in phosphate buffered saline, pH 7.2 and stained for ectodermal (βIII-tubulin), mesodermal (α-smooth muscle actin (α-SMA) or desmin) or endodermal (Sox17 or α-fetoprotein (α-SMA)) marker expression (Neely et al., 2011).

Statistical calculations

The standard deviation of the ratio of means of target gene (g) mRNA levels to control β-actin (c) mRNA levels was calculated as follows: μg/μc × √[(σg/μg)2 + (σc/μc)2] where μ = mean and σ = standard deviation (Wikipedia, 2013).

Results

Preparation and characterization of vector stocks

Vector stocks for each of the reprogramming vectors (and the mRFP1 vector) were prepared by transient transfection of HEK293T cells as described in Materials and Methods. The Yamanaka reprogramming vectors, however, do not express a marker gene that would enable easy determination of vector titer. Thus, most reprogramming experiments for generation of iPSCs use a ‘blind’ approach for transduction of target cells with an assumption of vector titer. We determined the titer of each vector stock preparation by isolation of genomic DNA of transduced HEK293T cells followed by qPCR using primers that targeted the packaging sequence in Moloney murine leukemia virus (Table 1). The PCR product size was 231 bp long and the optimal annealing and extension temperature was determined to be 63.1°C in a preliminary temperature gradient PCR experiment. For amplification of a control cellular gene, we chose human β-actin primers that gave a product size of 250 bp. The same annealing-extension temperature was used for β-actin amplification. Standards for qPCR were prepared by diluting a retroviral vector plasmid DNA in genomic DNA isolated from untransduced control cells. β-actin standards were generated by serial 10-fold dilution of genomic DNA. The results of vector titer determination are shown in Table 2.

Table 2 Titer determination of Moloney murine leukemia virus vectors encoding RFs or mRFP1 by qPCR.

Vector	Vector copy
number	β-actin copy
number	Vector titer
(IUa/ml)	
pMXs-hKLF4	6,830	60,000	569,167	
pMXs-hOCT3/4	2,530	48,500	260,825	
pMXs-hSOX2	2,830	42,100	336,105	
pMXs-h-cMYC	3,780	41,100	459,854	
pMXs-mRFP1	2,750	30,900	444,984	
Notes.

a IU, Infectious units.

Vector titer = (Vector copy number ÷ (β-Actin copy number ÷ 2)) × (number of cells used for infection) × (1,000 ÷ volume used for infection).

Number of cells used for infection = 250,000; Volume used for infection = 100 µl.

Table 3 Description of iPSC primers used in RT-qPCR.

Laboratory
designation	Description (nucleotide position)	Product size	Sequence (5′ – 3′)	Melt
temp	Parent GenBank
Accession No	
AT103	Human β-Actin, S (897-916)	84	CTGTGGCATCCACGAAACTA	59.7	NM_001101	
AT104	Human β-Actin, AS (961-980)	AGCACTGTGTTGGCGTACAG	60.0	
AT105	Human NANOG, S (347-366)	88	AGATGCCTCACACGGAGACT	59.9	NM_024865	
AT106	Human NANOG, AS (415-434)	TTGGGACTGGTGGAAGAATC	59.9	
AT75	Human DNMT3B, S (1394-1413)	203	ATAAGTCGAAGGTGCGTCGT	59.8	NM_006892	
AT76	Human DNMT3B AS (1577-1596)	GGCAACATCTGAAGCCATTT	60.1	
AT111	Human OCT4, S (750-771)	146	AAAGCGAACCAGTATCGAGAAC	59.8	NM_002701	
AT112	Human OCT4, AS (879-898)	GCCGGTTACAGAACCACACT	60.0	
AT175	Human OCT4, S 3′UTR (1243-1262)	55	AGGAAGGAATTGGGAACACA	59.4	NM_002701	
AT176	Human OCT4, AS 3′UTR (1279-1297)	AACCAGTTGCCCCAAACTC	60.0	
AT65	Human SOX2, S (1302-1320)	151	CCCAGCAGACTTCACATGT	57.4	NM_003106	
AT66	Human SOX2, AS 3′UTR (1433-1452)	CCTCCCATTTCCCTCGTTTT	57.8	
AT113	Human SOX2, S (632-651)	92	GATGCACAACTCGGAGATCA	59.8	NM_003106	
AT114	Human SOX2, AS (704-723)	GCTTAGCCTCGTCGATGAAC	60.0	
AT115	Human KLF4, S (2002-2021)	110	CACCTCGCCTTACACATGAA	59.7	NM_004235	
AT116	Human KFL4, AS 3′UTR (2092-2011)	CATCGGGAAGACAGTGTGAA	59.7	
AT117	Human KLF4, S(1774-1793)	67	GCCACCCACACTTGTGATTA	59.4	NM_004235	
AT118	Human KLF4, AS (1821-1840)	GTGCCTTGAGATGGGAACTC	59.7	
AT177	Human cMYC, S (1083-1102)	94	CTCCACCTCCAGCTTGTACC	59.7	NM_002467	
AT178	Human cMYC, AS (1157-1176)	GCTGTCGTTGAGAGGGTAGG	59.9	
AT179	Human cMYC, S 5′UTR (119-138)	74	AGGGATCGCGCTGAGTATAA	59.8	NM_002467	
AT180	Human cMYC, AS 5′UTR (173-192)	TGCCTCTCGCTGGAATTACT	60.0	
Notes.

S sense

AS antisense

Optimization of transduction of MRC-5 fibroblasts

MRC-5 fibroblasts were initially transduced with the mRFP1 encoding retroviral vector alone to determine optimal MOI and transduction parameters. We tested two different MOI (2.5 and 5.0), and transduction at unit gravity (1 × g) or spin-transduction at 1,200 × g for 2 h at room-temperature. The efficiency of transduction was estimated by fluorescence microscopy (Fig. 1). The majority of MRC-5 fibroblasts were transduced at both MOIs. The fluorescence intensity was higher in the spin-infection cultures than that observed at unit gravity, suggesting higher transduction efficiencies. This could not be categorically determined since the plate used for transduction at normal gravity (1 × g) seemed to have slightly lower light transmission characteristics than that used for transduction at 1,200 × g (as indicated by comparison of phase contrast and fluorescence images between the two plates, Fig. 1). However, parallel transductions carried out on primary human keratinocytes clearly revealed the higher efficiencies of transduction at 1,200 × g and with an MOI of 5 (Fig. S1). To ensure that the maximal numbers of cells were transduced, we chose an MOI of 5 with a spin-transduction protocol for reprogramming of MRC-5 fibroblasts. This also ensured that virtually all cells expressed the mRFP1 marker gene following transduction.

Figure 1 Spin-transduction of human fibroblasts with gammaretroviral vector is more efficient than transduction at unit gravity.

Human MRC-5 fibroblasts were transduced with pMXs-mRFP1 vector at unit gravity (1 × g) or by spin-transduction (1,200 × g for 2 h at room temperature). Two different MOIs were used (2.5 and 5.0). The cells were visualized by phase contrast and fluorescence microscopy.

Reprogramming of MRC-5 fibroblasts with Yamanaka vectors and detection of putative IPSC-like colonies based on differential mRFP1 expression

We combined appropriate volumes of the supernatants of the four reprogramming vectors and that of the mRFP1 encoding vectors to give an MOI of 5 for each vector and then pelleted the virus by ultracentrifugation. The pellet was resuspended in MRC-5 growth medium and used for transduction of the human fibroblast cells by two rounds of spin-transduction. The efficiency of transduction was determined as follows: (1) An aliquot of the transduced cells was analyzed by flow cytometry which indicated a transduction efficiency of 97% (Fig. 2A). (2) Photomicrographs of phase-contrast and fluorescence microscopy were obtained (Fig. 2B) and the number of cells in each was counted using the cell-counter feature of NIH ImageJ software. Approximately 95% appeared to be transduced by this approach. (3) Genomic DNA was isolated from untransduced and vector-transduced MRC5 and subjected to qPCR for quantitation of vector and β-actin copy numbers. From this, the average number of vector copies per cell was estimated. The results are shown in Fig. 8B and indicate that each cell contained around 25 copies of vector. This number is consistent with the use of an MOI of 5 for each of the five vectors in the transduction.

Figure 2 Efficiency of transduction of MRC-5 fibroblasts by RF vector cocktail.

(A) Untransduced and transduced MRC-5 fibroblasts (MRC-5 5V) were analyzed by flow cytometry and the percentage of transduced fibroblasts was determined after gating the untransduced cells to determine the ‘negative’ fraction. (B) Fluorescence and phase contrast images of untransduced and transduced MRC-5 cells. The percentage of transduced cells was determined using the cell-counter feature in NIH ImageJ software.

Following two rounds of transduction, and a brief period of expansion, the MRC-5 fibroblasts were plated on mouse ES feeder cells (SNL 76/7-4) (10,000 MRC-5 cells/well of 6-well plate), and the plates were observed on a regular basis in an inverted fluorescence microscope. The mRFP1 expressing MRC-5 cells exhibited a rapid expansion in culture as monitored by fluorescence microscopy. In the first two weeks after seeding on mouse feeder cells the colonies exhibited indistinct borders (Fig. 3). Between weeks 2 and 4, colonies with well-demarcated borders also appeared. Fluorescence microscopy revealed that the colonies with indistinct borders still expressed mRFP1, while the colonies with distinct borders were ‘dark’, i.e., did not express the marker gene (Fig. 4). Some colonies showed invasion by occasional mRFP1-positive fibroblast-like spindle shaped cells at their edges (Fig. 4). Phase-contrast microscopy revealed that the cells in mRFP1-negative colonies exhibited scant cytoplasm and had nuclei with prominent nucleoli that are characteristic of iPSCs (Fig. 4 and phase contrast image at 10× magnification on the last row of Fig. 4).

Figure 3 mRFP1-positive colonies and cell aggregates seen during reprogramming of human fibroblasts on mouse feeder cells.

Human MRC-5 fibroblasts transduced with four reprogramming Yamanaka retroviral vectors and pMXs-mRFP1 vector were seeded onto Mitomycin-C treated mouse feeder cells and cultured in human ES reprogramming medium containing bFGF. The resultant colonies and cell aggregates were visualized by phase contrast and fluorescence microscopy.

Figure 4 mRFP-1 negative putative IPSC colonies on mouse feeder cells.

Human MRC-5 fibroblasts transduced with four reprogramming Yamanaka retroviral vectors and pMXs-mRFP1 vector were seeded onto Mitomycin-C treated mouse feeder cells and cultured in human ES reprogramming medium containing bFGF. The resultant colonies were visualized by phase contrast or fluorescence microscopy. The bluish-black rings and stains seen at 4× magnification are the result of using an object marker to delineate putative IPSC colonies. (*) denotes a slight upward shift in field for the fluorescence image with respective to corresponding phase contrast image at 10× magnification in the top row. The images with (**) do not have corresponding 4× or 10× images and are therefore to be considered in pairs of phase contrast and fluorescence images.

We identified 24 colonies with iPSC-like morphology that were negative for mRFP1 expression. This provided a calculated efficiency of iPSC generation of 0.04% ((24 ÷ 60,000 seeded cells) × 100). Twenty-one of these were chosen for further cloning. Fourteen colonies survived picking and could be adapted to feeder-free conditions on Matrigel. The subcloned iPSCs continued to exhibit silencing of mRFP1 (Fig. 5). Six clones were subjected to a battery of tests to determine if they exhibited iPSC characteristics (Materials and Methods).

Figure 5 Cloned IPSC-like colonies retain mRFP1-negative phenotype.

mRFP1-negative colonies were selected and seeded onto fresh 6-well plates containing mouse feeder cells and visualized by phase contrast and fluorescent microscopy.

All clones expressed pluripotency markers alkaline phosphatase, SSEA3, SSEA4, Nanog, Oct4, and Tra-1-60 (Fig. 6A and Figs. S2 through S6).

Figure 6 Characterization of putative iPSC-5 clone.

(A) Parallel wells of the iPSC clone (iPSC-5) were fixed with paraformaldehyde and stained for SSEA4, Nanog, Tra-1-60 or SSEA3 and Oct4 pluripotency markers and visualized under fluorescence microscopy. The alkaline phosphatase stained colonies were visualized under bright field microscopy. (B) Karyotype of IPSC clone. (C) Embryoid bodies were allowed to attach and spread out on gelatin-coated wells, fixed and stained for endodermal (Desmin), mesodermal (α-smooth muscle actin (α-SMA)), or ectodermal (βIII-tubulin) markers.

Retroviral expression of RFs is shut off following successful reprogramming

To determine the relative levels of RFE (endogenous) and RFT (endogenous and vector-derived) mRNA expression, total RNA from iPSCs and control cells, were reverse transcribed and amplified by qPCR. The mRNA levels were normalized to β-actin levels. The normalized expression patterns are shown in Fig. S7. Untransduced control MRC-5 fibroblasts expressed low levels of OCT4 and SOX2 but readily detectable levels of KLF4 and cMYC. Transduced MRC-5 cells (MRC-5 5V), in contrast, exhibited high levels of RFT in comparison to RFE mRNA levels. All four RFs were raised in putative iPSCs. Counterintuitively, RFEs sometimes exceeded the RFTs for some RFs. To therefore control for possible differing efficiencies of amplification or reverse-transcription of RFT vs RFE mRNAs, we determined RFT to RFE ratio for all samples. We then normalized these ratios for each RF to that of hES cells in which the entire contribution to the RFT mRNA must originate only from endogenous sources. These normalized data are shown in Fig. 7A and indicate that the putative iPSC clones exhibited an average RFT/RFE ratio of the four RFs comparable to that in hES cells (with less than 2-fold difference). In contrast, vector-transduced MRC-5 cells, prior to reprogramming, exhibited 21- to 9,017-fold higher RFT/RFE ratios than hES or iPSCs. The vector-derived RF expression was then significantly decreased as the endogenous cellular transcription of the corresponding factors was increased. We also determined the expression of other genes characteristic of iPSCs (NANOG, DNMT3B). These were expressed at high levels in hES and the putative iPSC clones but not in untransduced MRC-5 or vector-transduced MRC-5 prior to reprogramming (MRV-5 5V) (Fig. 7B). These results indicate that the fibroblasts were fully reprogrammed.

Figure 7 Reprogramming factor and pluripotency related mRNA expression in putative iPSC clones.

(A) The RFT to RFE ratio, determined by RT-qPCR of total RNA isolated from indicated cell types, is shown for untransduced MRC-5 fibroblasts, vector-transduced MRC-5 fibroblasts (MRC-5 5V) and derived iPSC clones. The RFT/RFE ratios were normalized to that observed in hES cells. Mean ± standard deviation of the combined RFT/RFE ratios of all four RFs are shown above the bars, except for MRC-5 5V where the individual RFT/RFE ratio, while significantly different from hES, also varied significantly between the RFs. (B) Expression of NANOG and DNA methyltransferase 3B (DNMT3B) in hES cells, untransduced MRC-5 and transduced MRC-5 (MRC-5 5V), and derived iPSC clones. The mRNA expression was normalized to β-actin levels in the samples. Error bar represents one standard deviation. The standard deviation of the ratio of means was calculated as described under Materials and Methods.

Methylation status of 5′ LTR of Moloney vector in transduced MRC5 cells and derived iPSC clones correlates with gene expression studies

Silencing of expression from Moloney LTR promoter has been associated with methylation at CpG islands. To determine the methylation status of the viral LTR promoter, we designed an MSRE-qPCR assay that spans SmaI (methylation sensitive) and MscI (methylation insensitive) restriction sites in the 5′ LTR (Fig. 8A). The sense primer was located in the U3 region of LTR while the antisense primer was located in the 5′ UTR downstream of the primer binding site (PBS). This ensured that only the 5′ LTR was amplified in the qPCR assay. Genomic DNA from each iPSC clone was isolated and then digested with SmaI or MscI or mock-digested in the appropriate buffer. An aliquot of the digest was subjected to qPCR as described in Materials and Methods. For generating methylation controls, HindIII-linearized vector plasmid (4 µg) was methylated in vitro using CpG methyltransferase (M.Sss1; 4 units/µg of DNA, New England Biolabs, Massachusetts, USA) in the supplied buffer containing S-adenosylmethionine for 2 h at 37°C. The methylation reaction was terminated by heat inactivation at 65°C for 20 min. Genomic DNA from untransduced MRC5 cells was spiked with an aliquot of unmethylated or methylated vector plasmid and used as positive and negative controls for digestion and qPCR with SmaI or MscI. Genomic DNA from MRC5 fibroblasts transduced with all four reprogramming vectors as well as mRFP1 encoding vector was also digested with SmaI or MscI or undigested and then subjected to qPCR as for the iPSC clones. Each of the samples were also tested by qPCR for human β-actin as described in Materials and Methods.

Figure 8 Methylation analysis of Moloney vector 5′ LTR using MSRE-qPCR.

(A) Sequence of 5′ long terminal repeat (LTR) and untranslated region of Moloney murine leukemia virus. The U3, R and U5 sequences within the LTR are shown and demarcated by vertical lines. Also shown are direct repeats (DR1 and DR2), Tata box, polyadenylation signal (Poly A), negative control region (NCR), binding site for ELP/NR5A1, and primer binding site (PBS). The CpG nucleotides are marked underneath by ‘*’ to indicate putative sites of methylation. The methylation sensitive SmaI and methylation insensitive MscI restriction enzyme sites are shown in red and green, respectively. >>> and <<< identify forward and reverse primers used in MSRE-qPCR. (B) Genomic DNA from untransduced MRC-5, vector transduced MRC-5 cells (MRC-5 5V), iPSC clones, and MRC-5 control genomic DNA spiked with unmethylated (UPC) or methylated plasmid vector DNA (MPC) was either undigested (Uncut) or digested with SmaI or MscI and then subjected to qPCR for determination of vector 5′ LTR or β-Actin copy numbers. The vector copy numbers per cell was calculated as described in Materials and Methods. Error bars represent one standard deviation. The standard deviation of the ratio of means was calculated as described under Materials and Methods.

The results are shown in Fig. 8B and indicate that there were similar numbers of vector copies per cell for the parent transduced MRC-5 cells and the derived iPSC clones (22 to 25 copies/cell). In 5 of the 6 clones the number of vector copies per cell was identical between uncut and SmaI digested genomic DNA samples. The complete resistance to digestion by SmaI in these clones was consistent with methylation at the SmaI site. In clone 17, the genomic DNA was only partially resistant to SmaI. In contrast, in vector-transduced parent MRC-5 fibroblasts, the SmaI site was exquisitely sensitive to digestion, indicating that prior to reprogramming the site was not methylated. The in vitro methylated (MPC) and unmethylated vector plasmid (UPC) showed resistance and susceptibility to SmaI, respectively, as anticipated. In all cases, PCR amplification was significantly thwarted by predigestion with the methylation insensitive MscI in the same buffer as SmaI, indicating that the genomic DNA did not contain inhibitors to restriction enzyme digestion. These results corroborate the RT-qPCR results described above showing complete abrogation of vector derived gene expression in the tested iPSC clones. The results also explain the silencing of mRFP1 expression during reprogramming.

Further characterization of the clones (Figs. S2–S6) revealed that with the exception of clone #16 (that showed trisomy of chromosome number 12, Fig. S4B), all other clones exhibited a normal karyotype (Fig. 6B and Figs. S2–S5). EBs derived from these clones (Fig. 9) generated cells that expressed markers of ectoderm, endoderm, and mesoderm (Fig. 6C and Figs. S2–S6C) supporting their pluripotency.

Figure 9 Embryoid body derivation from an mRFP1-negative iPSC clone.

IPSCs growing in log phase were released with Dispase treatment and allowed to form embryoid bodies in low-attachment flasks as described in Materials and Methods. The flasks were observed periodically (at the indicated intervals) under the microscope and photographed.

Discussion

During reprogramming of somatic cells to iPSCs, the gene expression profile mimics that seen in embryonic stem cells and deviates from somatic cells, such as fibroblasts, from which the iPSCs originated. The shift in expression patterns occurs in stages (Stadtfeld et al., 2008) as outlined in the Introduction. One model (Liang & Zhang, 2013) posits that there are several barriers to overcome during the reprogramming of fibroblasts to iPSCs. The first barrier is during transition from mesenchymal to epithelial (MET) phenotype, the second barrier is for transition from epithelial to nascent iPSCs during which continued expression of RF is no longer required and can pose a barrier to the final transition to bona fide iPSCs. According to this model then, the down-regulation of expression from retroviral vectors is imperative to successful reprogramming to iPSCs. Cells that continue to express retrovirally derived RFs would be trapped in a partially reprogrammed state (Jaenisch & Young, 2008; Liang & Zhang, 2013; Okita, Ichisaka & Yamanaka, 2007). Hotta and coworkers classify partially reprogrammed and fully reprogrammed fibroblasts as Class I and Class II based on retention or extinction of retroviral expression, respectively (Hotta & Ellis, 2008). The Yamanaka group also correctly predicted that continued expression of RFs from the retroviral vectors would be antithetical to subsequent pluripotent differentiation of the iPSCs into other cell types (Takahashi & Yamanaka, 2006). Papapetrou and coworkers used lentivirus vectors encoding distinct fluorescent markers linked to each of the four RFs. They found that all four vectors were silenced in successfully reprogrammed iPSCs (Papapetrou et al., 2009).

Multiple studies have uncovered possible mechanisms for gene silencing of retroviruses in embryonic cells (Cherry et al., 2000; Minoguchi & Iba, 2008) (reviewed in Hotta & Ellis, 2008). Various silencers have been identified in the viral LTR. These include CpG islands in the promoter region (Hilberg et al., 1987), the proline t-RNA primer binging site (PBS) that is a target of Trim28 (Wolf & Goff, 2007), a negative control region containing binding site for the transcription factor YY-1, and an ELP/Nr5a1 binding site (Flanagan et al., 1989) (Fig. 8A). These sites allow the targeting of enzymes or proteins to the proviral LTR that mediate epigenetic modifications such as methylation of histones (H3K9 or H3K27) or DNA (i.e., methylation of cytosine in CpG islands) that ultimately result in silencing of the promoter (Pannell & Ellis, 2001). The finding of de novo upregulation of cellular methyltransferase (DNMT3B) expression in bona fide iPSCs could be the mechanistic link to this hypothesis (Stadtfeld et al., 2008).

The initial Yamanaka vectors were based on the pMXs backbone described by Kitamura and coworkers (Kitamura et al., 2003), as are the vectors used in this study. These vectors are susceptible to silencing during reprogramming. In contrast, other gamma-retroviral vectors that contain an altered primer binding site (glutamine t-RNA instead of proline t-RNA binding site) or those that contain the murine stem cell virus (MSCV) promoter are less prone for silencing as these vectors were specifically engineered for enhanced expression in stem cells (Hotta & Ellis, 2008).

Our finding that silencing of mRFP1 expression in reprogrammed iPSCs is consistent with these earlier reports. This contention is strongly supported by the loss of expression of RFs derived from retroviral vectors (Fig. 7A). In addition, we demonstrated an increase in DNMT3B expression in the iPSC clones (Fig. 7B) that is known to participate in DNA methylation resulting in silencing of gene expression. Finally, we demonstrated that the 5′ LTR promoter of the gammaretroviral vector copies in derived iPSCs were methylated at the SmaI site in U3 region by MSRE-qPCR (Fig. 8B).

In one instance, we identified a putative colony with iPSC characteristics with diminished mRFP1 expression as compared to surrounding unreprogrammed mRFP1 positive cells on day 7 of coculture on SNL mouse feeder layer. We carried out a live-cell staining of this colony after 19 days of coculture on SNL feeder layer using Alexa488-conjugated anti-Tra-1-60 antibody in conjunction with Hoechst 33342 nuclear staining as previously described. Photomicrographs of the colony (Fig. 11) revealed that the colony had lost the low level of mRFP1 noticed at 7 days of coculture and had acquired Tra-1-60 positive phenotype. In contrast, an mRFP1 strongly positive aggregation of cells did not exhibit Tra-1-60 positive phenotype (Fig. S8). The mRFP1 negative and Tra-1-60 positive colony was not characterized further. Nevertheless, this is additional evidence that reprogramming can be inferred from silencing of mRFP1 expression in transduced human fibroblasts.

Figure 10 Putative IPSC colony showing mRFP1-positive and mRPF1-negative cells.

mRFP-negative colonies were picked using a pulled glass Pasteur pipet as described in Materials and Methods and placed in a well with mouse feeder cells. The following day, the well was observed under phase contrast and fluorescence microscopy.

Figure 11 Live staining of mRFP1-negative colony for Tra-1-60 surface antigen.

A putative iPSC colony with variegated mRFP1 expression was identified on day 7 (dashed white circle). On day 19, the well was stained with Alexa488-labeled antibody to Tra-1-60, a pluripotency marker, and Hoechst 33342 for visualizing nuclei, and then observed by phase contrast and fluorescence microscopy as described in the text.

The putative iPSC clones were also validated using surface marker expression and differentiation into cell types of all three germ layers (Fig. 6C and Figs. S2–S6C). We chose not to determine pluripotency using teratoma formation, in line with the sentiments of other investigators who feel that validation of pluripotency of iPSCs by teratoma formation may not be necessary or even essential for demonstrating successful reprogramming of fibroblasts (Buta et al., 2013).

There are additional advantages to our strategy of marking primary cells with a fluorescent marker gene. The mRFP1-negative colonies were surrounded by mRFP1 positive unreprogrammed cells. This allowed us to monitor possible contamination of unreprogrammed cells during the colony picking process. An example of a colony fragment bearing unreprogrammed cells following the subcloning process is shown in Fig. 10. An alternative explanation could be variegated gene expression or incomplete reprogramming of this particular colony.

Other investigators have also engineered retroviral vectors to detect and select for successfully reprogrammed cells into iPSCs. Hotta and coworkers described the EOS lentiviral vector selection system for derivation of human iPSCs (Hotta et al., 2009a). This vector encodes enhanced green fluorescent proteins and puromycin resistance gene under control of a hybrid promoter consisting of the mouse early transposon in combination with enhancers derived from OCT4 and SOX2 promoters. The puromycin selection is linked to enhanced green fluorescent protein (EGFP) expression by means of an internal ribosomal entry site. Successful reprogramming results in enhanced expression of EGFP and allows the selection of these cells using puromycin. Alternative methods of selection of reprogramming include the use of selection markers under the FBX15 (Takahashi & Yamanaka, 2006) and NANOG (Okita, Ichisaka & Yamanaka, 2007) promoters. Selection of Fbx15 expression in mouse cells resulted in iPSCs that were not fully competent to form chimeras. In contrast, selection for NANOG promoter activation resulted in mouse iPSCs that shared features with mouse ES cells. The latter experiment required the use of BACs containing the NANOG promoter driving a selection marker. The above-described approaches use positive selection strategies. In contradistinction, our approach does not call for the use of a particular promoter to positively select for reprogrammed cells, which has a potential to introduce biases during the selection process. Warlich and coworkers (Warlich et al., 2011) employed a combined approach to identify reprogrammed cells. They used lentiviral vectors encoding codon-optimized RFs and dTomato fluorescent protein under control of spleen focus forming virus promoter to reprogram mouse cells that harbored an endogenous OCT4-green fluorescent protein reporter cassette. These investigators also found that successful reprogramming correlated with silencing of exogenous RF expression with simultaneous upregulation and expression of green fluorescent protein under control of the OCT4 promoter.

The Yamanaka group previously used gammaretroviral vectors encoding fluorescent marker to demonstrate that such vectors are indeed silenced during the reprogramming step (Nakagawa et al., 2008). Studies from the Hochedlinger group (Stadtfeld et al., 2008) defined the key stages that occur during reprogramming of fibroblasts into iPSCs. Chan and coworkers (Chan et al., 2009) using live-cell imaging to identify successfully reprogrammed cells based on proviral silencing and expression of TRA-1-60, DNMT3B and REX1. Our study then builds on these earlier findings in the following ways: (1) careful determination of vector titers using qPCR; (2) use of sufficiently high MOI and spin-transduction to ensure efficient marking of virtually all target cells with mRFP1; and (3) initial screening of putative iPSC clones primarily based on silencing of mRFP1 expression and colony morphology. These modifications lead to the successful and unambiguous derivation of human iPSCs from fibroblasts.

A possible disadvantage of our method is that high efficiencies of transduction with retroviral vectors are required to ensure that nearly all of the primary cells express the marker gene. For example, we were unable to use this approach for primary human keratinocytes due to the lower efficiencies of transduction (Fig. S1). If transduction efficiency is less than optimal, an additional step of sorting the transduced primary cells will be required for identification of successfully reprogrammed IPSC colonies based on gammaretroviral silencing.

In summary, we describe a method of marking primary fibroblasts during retroviral vector-mediated programming to facilitate detection and differentiation of reprogrammed from non-reprogrammed colonies. This methodology would be of value for investigators who are in the process of establishing iPSC technology in their laboratories as well as to monitor the effects of different reprogramming conditions, addition of growth factors, media changes, etc. We finally suggest that this technique may also prove useful for iPSC derivation using alternative approaches such as Sendai virus or transfection with episomal plasmids.

Supplemental Information

Figure S1 Spin-transduction of primary human kertinocytes with gammaretroviral vector is more efficient than transduction at unit gravity

Primary human keratinocytes (Invitrogen Corporation, USA, Catalog number 12332-001) were transduced with pMXs-mRFP1 vector at unit gravity (1 × g) or by spin-transduction (1,200 × g for 2 h at room temperature). Two different MOIs were used (2.5 and 5.0). The cells were visualized by phase contrast and fluorescence microscopy.

Click here for additional data file.

Figure S2 Characterization of putative IPSC-10 clone

(A) Parallel wells of iPSC-10 clone were fixed with paraformaldehyde and stained with antibodies to SSEA4, Nanog, Tra-1-60 or SSEA-3 and Oct3/4. Nuclei were stained using Hoechst 33342. Alkaline phosphatase expression was determined on unfixed cultures as described in the text. The stained colonies were visualized under fluorescence microscopy. (B) Karyotype analysis of clone. (C) Embryoid bodies were allowed to attach and spread out on gelatin-coated wells, fixed and stained for endodermal (α-fetoprotein (α-FP)), mesodermal (α-smooth muscle actin (α-SMA)), or ectodermal (βIII-Tubulin) markers.

Click here for additional data file.

Figure S3 Characterization of putative iPSC-15 clone

(A) Parallel wells of iPSC-15 clone were fixed with paraformaldehyde and stained with antibodies to SSEA4, Nanog, Tra-1-60 or SSEA-3 and Oct3/4 as described in the text. Nuclei were stained using Hoechst 33342. The stained colonies were visualized under fluorescence microscopy. The alkaline phosphatase stained colonies were visualized under bright field microscopy. (B) Karyotype analysis of clone. (C) Embryoid bodies were allowed to attach and spread out on gelatin-coated wells, fixed and stained for desmin, α-smooth muscle actin (α-SMA ) actin, or Sox17 differentiation markers.

Click here for additional data file.

Figure S4 Characterization of putative iPSC-16 clone

(A) Parallel wells of iPSC-16 clone were fixed with paraformaldehyde and stained with antibodies to SSEA4, Tra-1-60 or SSEA-3 and Oct3/4 or Nanog and visualized under fluorescence microscopy. Nuclei were stained using Hoechst 33342 (top two rows of photomicrographs) or Dapi (for Nanog counter stain). Alkaline phosphatase expression was determined on unfixed cultures as described in the text. (B) Karyotype analysis of clone. Arrow points to trisomy of chromosome 12. (C) Embryoid bodies were allowed to attach and spread out on gelatin coated wells, fixed and stained for endodermal (α-fetoprotein (α-FP)), mesodermal (α-smooth muscle actin (α-SMA)), or ectodermal (βIII-Tubulin) markers.

Click here for additional data file.

Figure S5 Characterization of putative iPSC-17 clone

(A) Parallel wells of iPSC-17 clone were fixed with paraformaldehyde and stained with antibodies to SSEA3, Nanog, Tra-1-60 or SSEA4 and Oct3/4. Nuclei were stained using Hoechst 33342. Alkaline phosphatase expression was determined on unfixed cultures as described in the text. The stained colonies were visualized under fluorescence microscopy. (B) Karyotype analysis of clone. (C) Embryoid bodies were allowed to attach and spread out on gelatin coated wells, fixed and stained for endodermal (α-fetoprotein (α-FP)), mesodermal (α-smooth muscle actin (α-SMA)), or ectodermal (βIII-Tubulin) markers.

Click here for additional data file.

Figure S6 Characterization of putative iPSC-18 clone

(A) Parallel wells of iPSC-18 clone were fixed with paraformaldehyde and stained with SSEA4, Nanog and Tra-1-60, or SSEA-3 and Oct3/4. Nuclei were stained with Hoechst. The colonies were visualized under fluorescence microscopy. The alkaline phosphatase stained colonies were visualized under bright field microscopy. (B) Karyotype of clone. (C) Embryoid bodies were allowed to attach and spread out on gelatin coated wells, fixed and stained for endodermal (α-fetoprotein (α-FP)), mesodermal (α-smooth muscle actin (α-SMA)), or ectodermal (βIII-Tubulin) markers.

Click here for additional data file.

Figure S7 RF and pluripotency related mRNA expression in putative iPSC clones by RT-qPCR

Expression of indicated plutipotency markers (NANOG), DNA methyltransferase 3B (DNMT3B) and RFs (OCT4, SOX2, KLF4 and cMYC) in hES cells, untransduced MRC-5 and transduced MRC-5 (MRC-5 5V), and derived iPSC clones. The mRNA expression was normalized to β-actin levels in the samples. Error bar represents one standard deviation. The standard deviation of the ratio of means was calculated as described under Materials and Methods.

Click here for additional data file.

Figure S8 Live staining of mRFP1-positive cellular aggregate for Tra-1-60 surface antigen

An mRFP1-positive cellular aggregate in the same well that contained the mRFP1-negative colony (Fig. 11) was visualized under phase and fluorescence microscopy as described in the text and legend to Fig. 11.

Click here for additional data file.

Additional Information and Declarations

Competing Interests

Author Contributions

The authors have no competing interests to declare.

Narasimhachar Srinivasakumar conceived and designed the experiments, performed the experiments, analyzed the data, contributed reagents/materials/analysis tools, wrote the paper.

Michail Zaboikin, Andrew M. Tidball and Asad A. Aboud conceived and designed the experiments, performed the experiments, analyzed the data, wrote the paper.

M. Diana Neely conceived and designed the experiments, performed the experiments, analyzed the data.

Kevin C. Ess conceived and designed the experiments, analyzed the data, contributed reagents/materials/analysis tools, wrote the paper.

Aaron B. Bowman conceived and designed the experiments, analyzed the data, contributed reagents/materials/analysis tools.

Friedrich G. Schuening conceived and designed the experiments, contributed reagents/materials/analysis tools.

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
