# Peer review of "Gammaretroviral vector encoding a fluorescent marker to facilitate detection of reprogrammed human fibroblasts during iPSC generation"

_PeerJ, doi:10.7717/peerj.224_

## Round 0.1 · original submission · Major Revisions

· Academic Editor

Major Revisions

The major points that need to be addressed int he revision are the inclusion of teratoma data to confirm pluripotency, the inclusion of an RFP control, and a more complete presentation of the qPCR data as suggested by one of the reviewers.

Reviewer 1 ·

Basic reporting

the paper conforms with standards and format of PeerJ

Experimental design

The experimental design is logical, where the authors have devised a new reprogramming vector and characterized its ability to make human iPSCs.

Validity of the findings

- the authors should provide in vivo teratomas to confirm pluripotency of their clones

·

Basic reporting

Srinivasakumar and collegues generate iPS cells using Yamanaka's vectors and add the pMXs-mRFP1 vector. Their rationale for doing so is that differentiating cells silence viral vectors. By this line of reasoning, any downregulated vector should be evidence of successful reprogramming. There are however a few major problems with this manuscript.
The experiments are aimed at establishing that the cells that downregulated RFP are IPSC, but the RFP positive controls are not present. Some of the dataset that demonstrate normal karyotype and multilineage differentiation are missing (S2-S7) and problems exist with the design and interpretation of the qPCR data showing expression of the reprogramming genes (Fig6)

In summary, some important technical details and some data are missing.

Experimental design

The manuscript fits the scope of the journal and clearly defines the research question. I do however feel that some experiments do not meet the technical standard for this type of work. The lack of description of the primers and experimental setup for the data displayed in figure 6 makes is difficult to interpret or reproduce the data displayed in that figure.

Validity of the findings

It is difficult to interpret the major claim of the manuscript, i.e. that the RFP marker facilitates detection of reprogrammed human fibroblast during iPSC generation, because the RFP positive control is not investigated. There is also no other information, for example using an OG2 reporter in combination with the RFP vector to confirm that the RFP downregulation is indeed a sign of achieving reprogramming in fibroblasts. The results concerning transduction efficiency seem to be either due to the use of inappropriate methods (fluorescence microscopy instead of flow cytometry) or could be solved by repeating the experiment with appropriate materials (plates that are equally translucent).

Additional comments

In the result section, (p6, line 181-), it is stated that the Yamanaka vectors do not contain marker genes, that do not allow direct vector titration. This is true for the Yamanaka vectors, but several groups (e.g. Zhang PloS One 2011, Warlich Molecular Therapy 2011) have already generated optimized vectors that 1) contain all reprogramming genes in one vector and 2) carry a marker gene, and often also Cre or Flp recombinase recognition sites, to allow excission of the reprogramming cassette.

In the materials section, it is mentioned that the vectors are separately generated and then pooled to give the appropriate MOI (line 96). I fail to see how transduction with a mix of 5 vectors would be better than the use of one reprogramming vector, also with regards of downregulation of the individual vectors.

The authors state that issues with the transmission characteristics of a plate that was used for 1200 x g transduction precluded comparison of transduction efficiencies. There are several problems with this part of the manuscript: 1) Surely, repeated measures should allow proper comparison of fluorescence intensity when transmission properties are matched before the experiment. 2) A better way of comparing fluorescence intensity is flow cytometry of the transduced cells. In addition, this method is more robust with respect to quantification. 3) qPCR Measurements of vector copy numbers (VCN) of each individual vector does not depend on fluorescence characteristics of the culture plates, and allows a comparison of vector copy numbers for each individual vector.

In line 226, cells are considered reprogrammed when they appear well-demarcated. The authors claim that these are putative IPSC-like colonies due to their 'dark' appearance in fluorescence microscopy. 1) Fluorescence microscopy is far less sensitive than confocal microscopy or flow cytometry, so a 'dark' appearance does not mean that the RFP is actually downregulated. Furthermore, retroviral vectors tend to undergo variegation, leading to different expression levels arising from the same vector between different cells, possibily due to the integration location of the vector. Even though the mRFP vector appears absent, this does not mean that all the other vectors are silenced as well. In fact, no information is given about the methylation state of the promoters of the vectors, neither is it shown that the RFP vector is still present in these cells.
2) The claim that these cells are IPSCs when the RFP vector is downregulated is only proven by the appearance of demarcated colonies. Other researchers use Nanog or Oct4 reporters to prove that the demarcated colonies switch on the appropriate transcription factors. In this manuscript it is unclear whether the cells are reprogrammed at the time the colonies appear demarcated. Only after the mRFP negative colonies are adapted to matrigel and characterized. Where no mRFP positive colonies treated similarly to show that it is indeed the RFP downregulation that is critical for IPSC formation?
Furthermore, the characteristization of the clones in S3-S7 is incomplete, since the embryoid body staining for desmin, SM actin, B3 tubulin and Sox17 fluorescence micrographs and karyograms are not shown for every clone.

The authors claim that the reprogramming vectors are shut down by analyzing the mRNA of endogenous origin and comparing that to total (cellular and vector derived) mRNA. They claim (line 244) that “In every instance endogenous expression was equal to or slightly greater than the total expression of the corresponding reprogramming vector”. When we look at figure 6, we indeed see that endogenous expression is usually greater than the total expression. How can a PCR that is designed to pick up endogenous or cellular and vector derived mRNA always give more signal than the endogenous signal alone. Shouldn't the total (endogenous + cellular) signal always be higher than the total signal? The data in this figure seems to contradict with the statement that RFP (and therefore the reprogramming vectors) are downregulated.
Is there a difference in PCR efficiencies between the PCRs? More information about the PCR (design, primers, efficiency) is required to clarify this point.


In Summary:
In my opinion, the major claim of the manuscript, concerning the use of RFP to determine whether a fibroblast is reprogrammed, based on the general tendency of reprogrammed cells to switch off their reprogramming vectors, is not demonstrated. In addition, the hypothesis that silencing of an additional marker vector with no relation to the reprogramming vectors is a key event during IPSC generation seems unlikely, which makes correct experimental design all the more important. The authors do not provide evidence, for example using available reporters, to show that their cells indeed are reprogrammed when RFP is downregulated. The qPCR data in figure 6 is difficult to interpret because data (primer design, PCR efficiency) is lacking.

---

## Round 0.2 · accepted · Accept

· Academic Editor

Accept

The revised manuscript has adequately addressed the concerns of the reviewers and is acceptable for publication, but one minor change should be included in the discussion:

One of the reviewers had pointed out the lack of teratoma data. Your rebuttal correctly points out that a number of researchers now believe that teratoma formation is not essential. However, your discussion references Buta et al and suggests that a consensus has been reached on this, which is not accurate since teratoma formation is still considered to be required by many stem cell biologists to prove pluripotency. Please rephrase it accordingly.

Your manuscript describes an important tool that will likely aid researchers in tracking individual cells undergoing reprogramming as well as contribute to the understanding of the silencing during reprogramming.

·

Basic reporting

no comments

Experimental design

no comments

Validity of the findings

With the addition of information about the transduction of keratinocytes, the FACS data and image analysis of transduced MRC, the qPCR data on Nanog and Dnmt3B as well as the methylation specific qPCR and vector copy number information together with the analysis of RFP positive and negative colonies with the Tra-1-60 stain, I feel this manuscript has improved on all parts I commented on and demonstrates that RFP can be used as an additional marker during iPSC generation.

The only remaining comment would be that the authors demonstrate 24 cases iPSC derivation using RFP downregulation. One might ask whether this number of colonies is sufficiently large, but since the paper clearly states that the RFP marker has to be used to aid iPSC identification,the data presented seem sufficient for this claim.